# Brief communication: Pancake ice floe size distribution during the winter expansion of the Antarctic marginal ice zone

Alberto Alberello[1,*], Miguel Onorato[2,3], Luke Bennetts[4], Marcello Vichi[5,6], Clare Eayrs[7], Keith MacHutchon[8], and Alessandro Toffoli[1]

[1]Department of Infrastructure Engineering, The University of Melbourne, Parkville, VIC 3010, Australia.
[2]Dipartimento di Fisica, Università di Torino, Torino, 10125, Italy.
[3]INFN, Sezione di Torino, Torino, 10125, Italy.
[4]School of Mathematical Sciences, University of Adelaide, Adelaide, SA 5005, Australia
[5]Department of Oceanography, University of Cape Town, Rondenbosch, 7701, South Africa.
[6]Marine Research Institute, University of Cape Town, Rondenbosch, 7701, South Africa.
[7]Center for Global Sea Level Change, New York University Abu Dhabi, Abu Dhabi, United Arab Emirates.
[8]Department of Civil Engineering, University of Cape Town, Rondenbosch, 7701, South Africa.
[*]now at: School of Mathematical Sciences, University of Adelaide, Adelaide, SA 5005, Australia

**Correspondence:** Alberto Alberello (alberto.alberello@outlook.com)

**Abstract.**

The size distribution of pancake ice floes is calculated from images acquired during a voyage to the Antarctic marginal ice zone in the winter expansion season. Results show that 50 % of the sea ice area is made up of floes with diameters 2.3–4 m. The floe size distribution shows two distinct slopes on either side of the 2.3–4 m range, neither of which conforms to a power law. Following a relevant recent study, it is conjectured that growth of pancakes from frazil forms the distribution of small floes ($D < 2.3$ m), and welding of pancakes forms the distribution of large floes ($D > 4$ m).

## 1 Introduction

Prognostic floe size distributions are being integrated into the next generation of large-scale sea ice models (Horvat and Tziperman, 2015; Zhang et al., 2015, 2016; Bennetts et al., 2017; Roach et al., 2018a). Early results show that the floe size distribution affects ice concentration and volume close to the ice edge, in the marginal ice zone, where ocean waves regulate floe sizes and floes are generally the smallest, meaning they are prone to melting in warmer seasons (Steele, 1992). However, at present the only field data available to validate and improve the models are empirical distributions derived for pack ice spanning several orders of magnitude (from few meters to tens of kilometres; e.g. Toyota et al., 2016) and none resolve floes below the meter scale.

Break up of pack ice often resembles a fractal behaviour similar to many brittle materials (Gherardi and Lagomarsino, 2015). It has been argued that exceedance probability of the characteristic floe size, $D$, expressed as number of floes, follows a power law $N(D) \propto D^{-\alpha}$, where the scaling exponent is $\alpha = 2$ if a fractal behaviour is assumed (Rothrock and Thorndike, 1984).

Most of the previous observations of the floe size distribution in the marginal ice zone (noting that no observations are in pancake ice conditions) conform to a truncated power law (Stern et al., 2018), with the $\alpha$ value varying among studies depending on season, distance from the ice edge and range of measured diameters. Some observations of floe size distributions have been interpreted using a split power law (e.g. Toyota et al., 2016), with a mild slope for smaller floes and a steeper one for larger floes. In most cases, the sharp change in slope is an artefact due to finite size effects (Stern et al., 2018), although in few instances the split power law behaviour might be consistent with the data (Stern et al., 2018). The truncated power law cannot explain two different slopes in the probability density function $n(D)$, suggesting that different mechanisms might in fact govern the distributions for small and large floes (Steer et al., 2008).

The power law behaviour has been verified for most cases but its universality has not been demonstrated yet (Horvat and Tziperman, 2017). Scaling parameters are typically estimated on the log-log plane with a least square fit, which leads to biased estimates of $\alpha$, and, as noted by Stern et al. (2018), without rigorous goodness-of-fit tests. In comparison, Herman et al. (2018) examined the size distribution of floes under the action of waves in controlled laboratory experiments, by analysing the probability density function $n(D)$, which revealed a fractal response due to an arbitrary strain (a power law) superimposed to a Gaussian break up process induced by the waves. The interplay of these mechanisms is hidden in the floe number exceedance probability.

Existing observations do not provide quantitative descriptions of the floe size distribution for pancake ice floes, which form from frazil ice under the continuous action of waves and thermodynamic freezing processes (Shen et al., 2004; Roach et al., 2018b). This is important, for example, during the Antarctic winter sea ice expansion, when hundreds of kilometres of ice cover around the Antarctic continent is composed of pancake floes of roughly circular shape and characteristic diameters 0.3–3 m (Worby et al., 2008). Pancake floes represent most of the Antarctic sea ice annual mass budget (Wadhams et al., 2018). Moreover, in the Arctic, pancakes are becoming more frequent than in the past due to the increased wave intensity associated with the ice retreat (Wadhams et al., 2018; Roach et al., 2018b).

Shen and Ackley (1991) reported pancake floe sizes from aerial observations collected during the Winter Weddell Sea Project (July 1986), showing that pancake sizes increase with distance from the ice edge, from 0.1 m in the first 50 km up to $\approx 1$ m within 150 km from the edge (but without investigating the floe size distribution). They attributed this to the dissipation of wave energy with distance into the ice-covered ocean, and proposed a relationship between wave characteristics, mechanical ice properties and pancake size (Shen et al., 2004). More recently, Roach et al. (2018b) used camera images acquired from SWIFT buoys deployed in the Beaufort Sea (Sea State cruise, October–November 2015) to quantify the lateral growth of pancakes and their welding. A correlation between wave properties and the size of relatively small pancakes (up to 0.35 m) was confirmed.

To our knowledge, the pancake floe size distribution has yet to be characterised, noting that although Parmiggiani et al. (2017) developed an algorithm for pancake floes detection, they did not provide quantitative indication on the shape and size of

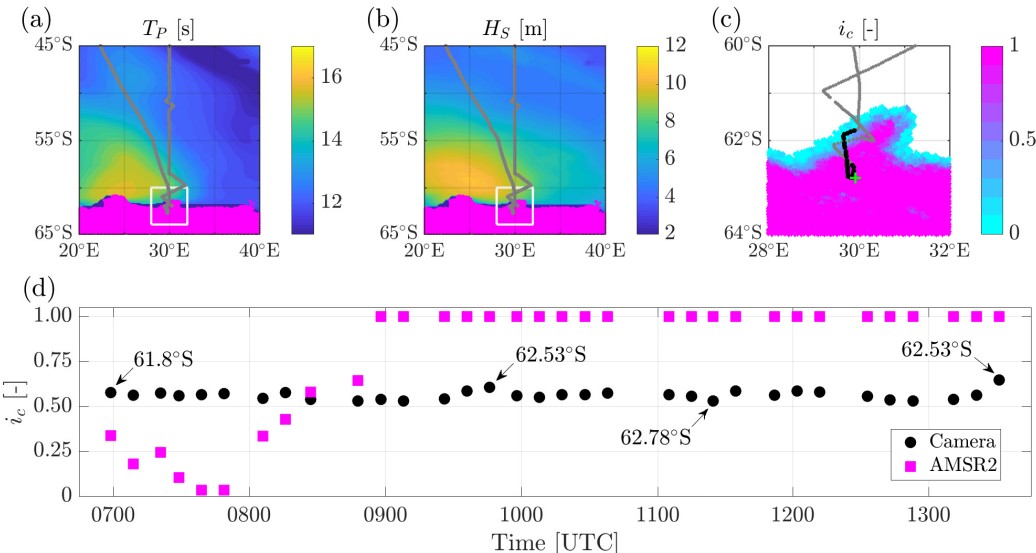

**Figure 1.** Environmental conditions during on the 4th of July 2017 (local time UTC+2). Peak wave period (a) and significant wave height (b) are sourced from ECMWF ERA-Interim reanalysis. The magenta area denotes ice and grey dots show the ship track. In (c), which is the subdomain indicated by a white frame in (a) and (b), ice concentration is sourced from the AMSR2 satellite with a 3.125 km resolution (Beitsch et al., 2014). The black dots denote the position during which cameras were operational and measurements undertaken. The green cross the location of deployment of a wave buoy. In (d), pancake floe concentration reconstructed from the camera images is shown as black dots, and total ice concentration obtained from AMSR2 satellite at the location closest to the measurements is shown as magenta squares.

the floes. Here, a new set of images from the Antarctic marginal ice zone are used to measure the shape of individual pancakes and to infer their size distribution.

## 2 Sea ice image acquisition

At approximately 07:00 UTC on the 4th of July 2017, the icebreaker S.A. Agulhas II entered the marginal ice zone between
5  61° and 63° South and approximately 30° East during an intense storm (see Fig. 1a,b for the ship track and a snapshot of peak wave period and significant wave height as sourced from ECMWF ERA-Interim reanalysis, Dee et al. 2011). A buoy was deployed in the marginal ice zone $\approx 100\,\mathrm{km}$ from the ice edge (green mark in Fig. 1c). At the time of deployment, the significant wave height was 5.5 m, with maximum individual wave height of 12.3 m. The dominant wave period was 15 s.

A system of two GigE monochrome industrial CMOS cameras with a 2/3 inch sensor was installed on the monkey bridge
10  of the icebreaker to monitor the ocean surface. The cameras were equipped with 5 mm C mount lenses (maximum aperture $f/1.8$) to provide a field of view of approximately 90°. The cameras were installed at an elevation of $\approx 34\,\mathrm{m}$ from the waterline and with their axes inclined at 20° with respect to the horizon. The system was operated by a laptop computer. Images were

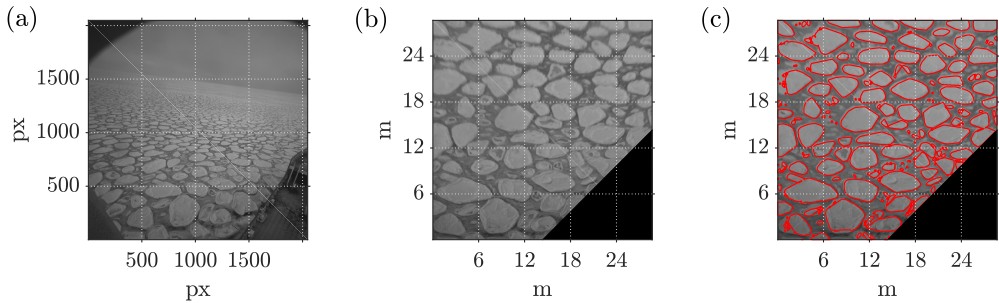

**Figure 2.** Sample acquired image (a), rectified and calibrated image (b) and detected pancakes (c).

recorded with resolution of 2448×2048 pixels and a sampling rate of 2 Hz during daylight on the 4th of July (from 07:00 to 13:30 UTC).

An automatic algorithm was developed using the MatLab Image Processing Toolbox (Kong and Rosenfeld, 1996) to extract sea ice metrics from the recorded images (see Fig. 2a for an example). To ensure statistical independence of the data set (i.e. to avoid sampling the same floe twice), only one camera and one image every 10 s was selected for processing (this interval guarantees no overlap between consecutive images). Images were rectified to correct for camera distortion and to project them on a common horizontal plane. A pixel to meter conversion was applied by imposing camera-dependent calibration coefficients. The resulting field of view is 28 m×28 m and resolution 29 px/m (see Fig. 2b). The image was processed to eliminate the vessel from the field of view, adjust the image contrast, and convert the grey scales into a binary map based on a user selected threshold. The mapping isolates the solid ice shapes from background water or frazil ice. The binary images, however, are noisy and require refining based on morphological image processing to improve the fidelity of the shape of identified pancake floes (i.e. erosion, filling and expansion). Threshold selection and morphological operations are optimised to detect pancake floes only and exclude interstitial frazil ice. (The optimisation is performed for the specific light and ice conditions using this particular camera setup.) The resulting separated floes are shown in Fig. 2c. Post-processed images were visually inspected for quality control, and $\approx 5\%$ of the images were discarded due to unsatisfactory reconstruction of the pancakes. Macroscopic differences between the acquired image and the reconstructed floes were noted, e.g. multiple floes were detected as one (artificial welding) or individual floes were divided into multiple floes by the automatic algorithm.

Identification of individual pancakes allows estimation of the individual floe areas $S$. An overall ice concentration ($i_c$, Fig. 1d) can be computed as the ratio of the area covered by pancake floes to the total surface in the field of view. A representative concentration was estimated every 60 consecutive images (i.e. 10 min time window), which is equivalent to a sampled area of 0.047 km$^2$. Pancake concentration was consistently $\approx 60\%$ with no significant variations throughout the day (Fig. 1d). The observed pancake concentration diverged from satellite observations (AMSR2) of sea ice concentration (see Fig. 1d), as the AMSR2 concentration includes the interstitial frazil ice, which is intentionally excluded from the image processing (i.e. detection of pancake ice only). Moreover, satellite data are an average over two daily swaths. Due to the intense storm activity and the associated drift of the ice edge ($\approx 100$ km Eastward in a day) at that time, this average may not be fully representa-

tive of the instantaneous conditions, resulting in an under- or over-estimation of the in situ ice concentration. In this regard, bridge observations following the Antarctic Sea Ice Processes and Climate protocol (ASPeCt, Worby et al., 2008), indicated a 90–100% concentration of total ice, where pancake ice was the primary ice type with concentration of 50–60% for most of the cruise (de Jong et al., 2018), in agreement with the image processing.

## 3   Pancake ice shape and floe size distribution

Approximating the floe shape as an ellipse, major ($D_1$) and a minor ($D_2$) axes are extracted. It is common practice, however, to define one representative dimension as a characteristic diameter $D = \sqrt{4S/\pi}$, by assuming that the pancake is a disk (Toyota et al., 2016), noting that other metrics are also widely used, e.g. the mean caliper diameter (Rothrock and Thorndike, 1984). Only floes entirely within the field of view are considered for these operations. Detection of small floes with $D < 0.25$ m is prone to error due to the limited number of pixels of which these floes are comprised and, thus, excluded from the analysis (Toyota et al., 2011). Moreover, a small fraction of large floes ($< 10\%$ of floes larger than 5 m) were artificially welded by the image processing. These floes were also excluded. In total, $4 \times 10^5$ individual floes were considered over an equivalent sampled area of $\approx 1.55\,\mathrm{km}^2$, and spanning almost 100 km of non-contiguous marginal ice zone.

Fig. 3a presents a scatter plot of the aspect ratio ($D_1 : D_2$). On average $D_1$ is $\approx 60\%$ greater than $D_2$ (slope of a linear fit). This aspect ratio is similar to one observed for broken ice floes (Toyota et al., 2011). The inset shows the full probability distribution of the ratio $D_1/D_2$ and indicates that floes elongated such that $D_1/D_2 > 3$ are infrequent. Fig. 3d shows the circularity $C = 4\pi S/P^2$, where $P$ is the floe perimeter (for a circle $C = 1$), which characterises the shape of the floes, noting that other metrics can be used to define the roundness of the floes (Hwang et al., 2017). For floes up to $D \approx 6$ m, the average circularity, denoted by the continuous line, is $C \geq 0.75$. Similar values have been reported for much larger broken floes (Lu et al., 2008).

Fig. 3b and 3e display the floe size area distribution as exceedance probability and probability density function respectively. Fig. 3e shows that, in terms of the equivalent diameter ($D$), 50% of the pancake area is comprised of floes with diameters in the range 2.3–4 m. The mode of the area distribution is 3.1 m (median and mean are $\approx 3.1$ m and $\approx 3.2$ m respectively), compared to $D_1 = 4$ m and $D_2 = 2.6$ m using the major and the minor axes.

Fig. 3c shows the exceedance probability $N(D)$, which exhibits two distinct slopes in the log-log plot, with a smooth transition from mild to steep slopes around the dominant diameter of 3.1 m. The probability density function of the equivalent diameter $n(D)$, shown in Fig. 3f, displays a pronounced hump in the transition between these regimes, revealing a third regime (2.3 m$< D <$ 4 m) around the modal pancake diameter, which is hidden in the exceedance probability, where the small- and large-floe regimes are defined as $D < 2.3$ m and $D > 4$ m (somewhat arbitrarily).

Small floes ($D < 2.3$ m) constitute the vast majority of the total detected floes ($> 80\%$). In this regime, the mild slope of $N(D)$ may result from a continuous process of floes accretion (from frazil to larger pancakes) regulated predominantly by thermodynamic freezing processes (Roach et al., 2018b). Floes larger than 4 m are detected far less frequently ($< 5\%$ of the total floes), and the steeper slope indicates that their size is most likely governed by different underlying physical mechanisms.

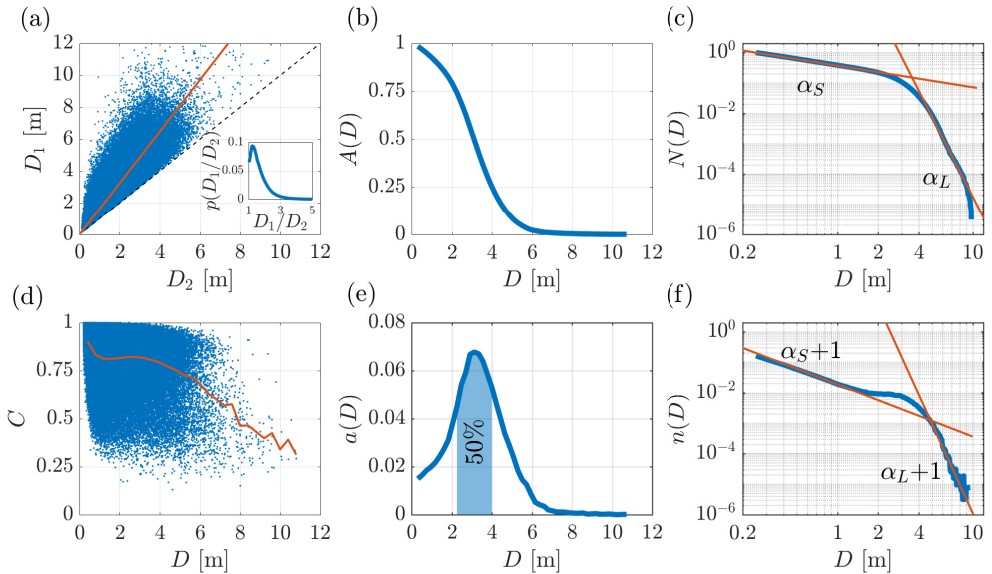

**Figure 3.** In (a), scatter plot of the major and minor axis of the pancake floe with the linear fit (solid orange line), the inset shows the probability density function of $D_1/D_2$. In (b), ice area distribution as a function of the floe diameter expressed as exceedance probability. In (c), floe number exceedance probability $N(D)$ as a function of the floe diameter with two power laws (solid orange lines) fitted for small ($D < 2.3$ m) and large floes ($D > 4$ m) respectively. In (d), scatter plot of the circularity of the floes against the equivalent diameter and the average value (solid orange line). In (e), ice area distribution as a function of the floe diameter expressed as probability density function. In (f), floe number probability density function $n(D)$ as a function of the floe diameter with two power laws (solid orange lines) fitted for small ($D < 2.3$ m) and large floes ($D > 4$ m) respectively.

Visual examination of the acquired images shows that the majority of the large floes are composed of two or more welded pancakes suggesting that the welding process, promoted by the high concentration of pancakes and the presence of interstitial frazil ice (Roach et al., 2018b), could be the dominant underlying mechanism for the shape of the probability distribution of large floes. Finite size effects are ruled out because the change in slope occurs for $D \approx 4$ m which is considerably smaller than

5   the image footprint.

Assuming, as standard, a power law $N(D) \propto D^{-\alpha}$ as a benchmark and using the maximum likelihood method following Stern et al. (2018), we determine $\alpha = \alpha_S = 1.1$ for small floes ($D < 2.3$ m) and $\alpha = \alpha_L = 9.4$ for large floes ($D > 4$ m). (Note that the maximum recorded diameter was $D = 10.8$ m, and, therefore, the estimation of the scaling exponent is not particularly meaningful or robust in either of the two regimes, as less than a decade of length scales are available.)

10   The power-law fits are approximations only, and an objective Kolmogorov–Smirnov goodness-of-fit test (Clauset et al., 2009) reveals that the empirical pancake size distribution does not scale accordingly to a power law in either the small- or large-floe regime, noting the power law hypothesis is more likely to be rejected when tested over limited diameter ranges (i.e. less than a decade). A close inspection of the empirical distribution shows that $N(D)$ possesses a slightly concave-down curvature across

all the diameter ranges (in a log-log plane), which is commonly associated with a truncated power law (Stern et al., 2018). The corresponding $n(D)$ displays an S-shape in the small-floe regime (it shifts from a concave-down to a concave-up curvature at $D \approx 1\,\text{m}$) in contrast to the hypothesis of a power law behaviour. Deviations from the power law scaling are prominent towards the extremes of the intervals ($D \to 0.25\,\text{m}$ and $D \to 2.3\,\text{m}$ for the small-floe regime; $D \to 4\,\text{m}$ and $D \to 10\,\text{m}$ for the large-floe

regime) but become conspicuous only by examining the empirical distribution over limited diameter ranges and probability intervals (i.e. zooming in on Figs. 3c–f). We also note that the increasing $a(D)$ in the small-floe regime (Figs. 3e) is inconsistent with a power law for $\alpha_S \geq 1$, as the area and number distributions are proportional to each other, i.e. $a(D) \propto D^2 n(D)$. Values of $\alpha_S \geq 1$ may be because the exponent has been estimated over a range of less of decade of diameters making its estimation non-robust. The discrepancy between area and number distribution confirms that the underlying number distribution is not a

power law, although we note that $\alpha_S \in (0.9, 1)$ provides a qualitatively good fit for the number distribution and is consistent with growing area distribution.

Goodness-of-fit tests also rule out floe size distributions such as the truncated power law (Stern et al., 2018), generalized Pareto (Herman, 2010), and linear combination of Gaussian distribution and power law (Herman et al., 2018). It appears that an accurate approximation of the floe size distribution (in the goodness-of-fit sense) can only be achieved by dropping any a priori

assumptions on the functional shape, e.g. by using a nonparametric kernel density estimation (Botev et al., 2010). However, this does not provide any insight on the underlying physical processes responsible for the shape of the empirical distribution.

## 4   Conclusions

Observations of pancake ice floe sizes during the winter expansion of the Antarctic marginal ice zone were analysed. An automatic floe detection algorithm was used to extract metrics (diameter and area) of the pancake floes, for which the equivalent

diameter ($D = \sqrt{4S/\pi}$) ranged between 0.25–10 m. This allowed a quantitative representation of the pancake size distribution to be discussed.

The floe size distribution displays three distinct regimes, which are visible in the probability density function that, compared to the commonly reported exceedance probability, is more informative. One regime is $D = 2.3$–4 m, centred around the dominant pancake diameter of 3.1 m, which covers half of the total pancake area, and appears as a hump in the probability

density function. Two different behaviours are observed for smaller and larger pancakes on a log-log plane. The small-floe regime ($D < 2.3\,\text{m}$), in which it is conjectured that pancakes are experiencing thermodynamic growth, is characterised by a mild negative slope (in terms of the floe number exceedance and probability density function), while the large-floe regime, in which floes are typically formed by welding (detected from visual analysis), is characterised by a much steeper slope noting that neither of the two regimes conform to a power law scaling.

These results reflect observations collected under storm conditions and, thus, lack generality. Simultaneous measurements of waves, floe size and heat fluxes under a number of different conditions are needed to verify the conjecture that different physical mechanisms (e.g. thermodynamic growth and welding) are responsible for the peculiar shape of the pancake ice floe size distribution.

*Code and data availability.* The detection algorithm and the acquired images are available upon request to the corresponding author.

*Competing interests.* The authors declare that they have no conflict of interest.

**Appendix A: Pancake detection algorithm**

The algorithm for the pancake detection is developed using the MatLab Image Processing Toolbox and built–in functions.

1. *Rectification*: projects the distorted camera image on an horizontal plane based on the camera internal parameters and the angle of view;

2. *Contrast adjustment*: contrast in the greyscale image is enhanced based on a CLAHE algorithm (the limit for clipping and shape of the distribution are user selected) to better isolate the pancakes from the frazil ice;

3. *Masking*: removes the ship from the field of view;

4. *Binary conversion*: the greyscale image is converted into a binary image where 1 corresponds to white (i.e. ice) and 0 to water or frazil (the threshold for conversion is user selected);

5. *Cleaning*: this morphological operation removes isolated white pixels (i.e. 1s completely surrounded by 0s);

6. *Erosion*: this morphological operation helps to separate the blobs corresponding to the pancakes (the erosion value is user selected);

7. *Filling*: this morphological operation substitute 0s with 1s in area completely enclosed by white pixels;

8. *Dilatation*: this morphological operation counterbalance the ice pixels lost by the erosion without merging two separate blobs;

9. *Clear border*: removes blobs intersecting the border of the field of view;

10. *Labelling and properties extraction*: geometrical properties of each individual floe are extracted.

All thresholds are user selected and the parameters have been subjected to testing to find the combination of operations that provided the best reconstruction as evaluated by the user visual inspection.

*Acknowledgements.* The cruise was funded by the South African National Antarctic Programme through the National Research Foundation. This work was motivated by the Antarctic Circumnavigation Expedition (ACE) and partially funded by the ACE Foundation and Ferring Pharmaceuticals. Support from the Australian Antarctic Science Program (project 4434) is acknowledged. MO was supported by the "De-
25 partments of Excellence 2018–2022" Grant awarded by the Italian Ministry of Education, University and Research (MIUR) (L.232/2016).

CE was supported under NYUAD Center for global Sea Level Change project G1204. The authors thank Lotfi Aouf at Meteo France for providing reanalysis data and the editor Ted Maksym for useful comments. AA and AT acknowledge support from the Air-Sea-Ice Lab Project. MO acknowledges B GiuliNico for interesting discussions. AA, AT and MO thank LE Fascette for technical support during the cruise.

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
