# Peer review of "Brief communication: Pancake ice floe size distribution during the winter expansion of the Antarctic marginal ice zone"

_The Cryosphere, 2018_

## Referee Comment (RC1) · C. Horvat (Referee) · 13 Aug 2018

This manuscript uses shipboard camera images taken over a transect of the Antarctic MIZ to describe the pancake floe size distribution in that region. The manuscript is sets out its goals, and accomplishes them concisely and straightforwardly, and so I recommend its publication in short order: this information is valuable and interesting to those who are trying to evaluate and understand sea ice models that incorporate the physics of small floes.

I do have some relatively minor issues, mainly related to presentation and data processing, that I would like to see improved upon before publication. These are listed

below - if any comments are unclear please contact me with questions.

Line 10 - I believe the paper of yours truly you mean to cite is H+T 2015, as that is the model paper.

Line 11 - While it is true early results show the importance of the FSD at the edge, only floe breaking by waves and floe melting has been put in models, so it isn't true that this is the place where floes are most important as we don't have a handle on their evolution deeper into the pack.

Line 12 - Please cite either Steele, 1992, or Horvat et al, 2016 when making the statement about floe melting.

Line 13 - I would delete "formed ... currents" as you also mention the importance of waves for pancake formation, and winds and currents can significantly alter the formation mechanism of sea ice in the MIZ.

Line 15 - change "exhibits" to "resembles" as you argue below the inapplicability of fractal scaling.

Line 3 - add citations to Herman 2014, 2017 here as it is important for readers to know there is really not much evidence of a multi-decadal power law.

Line 5 - Both papers cited here really argue *against* the adoption of FSD power laws, not for them!

Line 16 - "the *Antarctic* sea ice annual mass budget"

Line 18 - I'm not sure the statement about pancakes being more common is supported by the Roach et al paper as it is a point measurement.

Fig 1 - I would like to see this visualization improved, and the caption more descriptive. For example, it isn't clear that (a-b) and (c) are on different axes immediately, and isn't mentioned that (c) is the cutout in (a-b). Also, are the measurements you are making the dark line in (c)? Could you add the green dash to (a-b) as well?

Line 8 - explain what you mean about statistical independence.

Line 8-15 - generally, please explain the operation used to compute the pancakes as this is very important information for reproducing or building from this work.

Line 5 - please explain why one might decline an image

Line 8 - you explain why AMSR might *overestimate* concentration, but for many points along the track it underestimates concentration - explain.

Line 18 - "prone to error" - what kind of error? why?

Line 19 - why were the welded floes excluded? Isn't this the process by which these pancakes are said to form? What criteria is used to pick floes to exclude, and how does this affect the tail of your distributions in Fig 3?

Line 20 - area = 1.55 km^2 - I thought the swath was 28 m, which would mean you traveled 55 km into the ice, not close to 100 km.

Figure 3 - You have extra space in this figure - could you please also plot the area-weighted FSD rather than the number size distribution? The area size distribution is what appears in the Roach et al model and so would be good to see.

You can estimate how much spread there is in (a) by taking $D_1$ and adding white noise to it and calling that $D_2$, then re-sorting in the instances $D_2 > D_1$. the magnitude of the white noise that is required to get the fit line would tell you how much error there is

in assuming a circle.

Line 2 - why the mode = peak probability? Why not the mean, and could you report the "roundness" of the floes?

Line 4 - That the probability of exceedence hides the fact of a non-power-law distribution is extremely interesting and while some have discussed this in model papers, to my knowledge this is the first time this has been evidenced *on purpose* in an observational paper. I would like to see this highlighted!

Line 12 - I don't think you can say that there is a different physical mechanism to make larger floes as you only have a point observation of welding.

Line 25 - The point about dropping a priori assumptions is good, I would add that using a KDE is in some ways equivalent to fitting distributions to power laws: both are not derived from first principles and so both give little insight into the actual physics governing the distribution.

---

## Referee Comment (RC2) · Anonymous Referee #2 · 1 Sep 2018

The authors acquired images of pancake ice floes from a ship-based camera on July 4, 2017, in the Antarctic marginal ice zone at about 30E, 62S. An automatic algorithm identified pancake ice floes in the images, whose size distribution was then plotted. They found three size regimes: diameters 0.25 to 2.3 meters, 2.3 to 4 meters, and 4 to 10 meters (see Figures 3b and 3d). They conjecture that the small regime is driven by the growth of pancakes from frazil ice, and the large regime is driven by the welding together of pancakes.

This paper is basically a report of data analysis. The conjectures regarding the small and large floe regimes are just that – conjectures – without supporting evidence. I

have questions about the analysis, detailed below, as well as other comments. In my opinion, this paper needs major revisions.

I would like to note that I have not read any of the comments posted on the discussion page that accompanies this paper, so this is a completely independent review.

Comments in page order

Page 1, lines 9-10. The floe size distribution (FSD) was first integrated into a sea-ice model by Zhang et al in 2015 and 2016:

Zhang, J, Schweiger, A, Steele, M and Stern, H. 2015. Sea ice floe size distribution in the marginal ice zone: Theory and numerical experiments. J Geophys Res 120. DOI: https://doi.org/10.1002/2015JC010770

Zhang, J, Stern, H, Hwang, B, Schweiger, A, Steele, M, Stark, M and Graber, H. 2016. Modeling the seasonal evolution of the Arctic sea ice floe size distribution. Elem Sci Anth 4. DOI: https://doi.org/10.12952/journal.elementa.000126

Page 1, lines 12-15. The authors imply that there is not much "field data" available on floe sizes. I assume this refers to in-situ data such as that acquired from a ship in the ice. But there is plenty of remote sensing data, and it's not clear to me that field data is any better than remote sensing data, so the lack of field data does not seem like a shortcoming. The only advantage I can see to field data is the higher spatial resolution – in this case, the ability to identify floes as small as 0.25 m in diameter. But this advantage is not mentioned by the authors. Perhaps one of the values of this study is that it identifies floes that are smaller than in any other study. The fact that it consists of data collected in the field is not in itself a selling point, in my opinion.

Page 2, lines 2-4. This is really an oversimplification. It is certainly true that observations do not support a unique scaling exponent of the FSD – that was the subject of an entire paper (Stern et al 2018) which is cited by the authors later but not here. Only some of the 18 studies examined in that paper report "two distinct scaling exponents".

The authors imply that those exponents are given by Toyota et al 2011, but other studies have found different exponents over different ranges, and some have reported that a single exponent characterizes the FSD.

Page 2, lines 5-6. "The validity of power law scaling has not been demonstrated yet and its adoption is mostly justified by the wide range of floes diameters". Actually the validity of power-law scaling has been demonstrated in some cases, and I have never seen a paper that claims that power-law scaling is justified by the wide range of floe diameters. The papers cited by the authors don't make that claim.

Page 2, lines 6-7. "Scaling parameters are typically estimated on the log-log plane with a least square fit" – it would be good to note that such a procedure leads to a biased estimate of the scaling parameter.

Page 2, line 12. "Existing observations do not provide quantitative descriptions of the floe size distribution for pancake ice floes", but line 19 says "Shen and Ackley (1991) reported pancake floe sizes..." so doesn't that contradict line 12?

Page 2, line 26. "To our knowledge, the pancake floe size distribution has yet to be characterized." Take a look at: Parmiggiani, Moctezuma-Flores, and Guerrieri, Identifying pancake ice and computing pancake size distribution in aerial photographs, Proc. SPIE 10422, Remote Sensing of the Ocean, Sea Ice, Coastal Waters, and Large Water Regions 2017, 104220K (13 October 2017); doi: 10.1117/12.2277537 http://spie.org/Publications/Proceedings/Paper/10.1117/12.2277537

Section 2, Sea ice image acquisition. All good, nice work. I do have one comment: page 3 line 14 says that morphological image processing was used "to improve the shape of the pancake floes." I don't think "improve" is the right word! How about "to smooth"?

Page 4, lines 15-17. There is more than one way to define the floe diameter D. The first study of the FSD, Rothrock and Thorndike 1984, used the mean caliper diameter.

Page 5, line 5. "a power law N(D) proportional to Dˆ-alpha as a benchmark and using the maximum likelihood method". Here, N(D) is the cumulative distribution function (CDF), but the maximum likelihood method yields the best-fitting exponent of the probability density function (PDF), not the CDF. If the authors used the maximum likelihood method to obtain the best-fitting exponent of the PDF (call it -beta), then they would have had to convert it to the exponent of the CDF (via -alpha = -beta + 1). Did they do this?

Page 5, lines 6-8. The authors note that the range of floe sizes for the large regime (4 to 10 meters) spans less than a factor of 10, and therefore "the estimation of the scaling exponent for D > 4 m is rigorously not applicable". Well, the same is true for the small floe regime (0.25 to 2.3 meters) – it spans less than a factor of 10, so apparently the estimation of the scaling exponent for D < 2.3 m is rigorously not applicable either. Doesn't that destroy the basis of the floe size analysis here?

Page 5, lines 12-13. "the steeper slope indicates that their size is governed by different underlying physical mechanisms." Or by the finite size effect, in which larger floes are under-observed because the finite size of the images makes it less likely to see larger floes in their entirety. This has been described in the literature. Can the finite size effect be ruled out here?

Page 5, Figure 3. It looks to me like Figure 3b (the area distribution, a(D)) is not compatible with Figure 3d (the PDF, n(D)). In 3b, the area distribution increases as D increases, from D=0.2 to D=3. In that range, alpha_S = 1.1 so alpha_S + 1 = 2.1 so the PDF n(D) scales like Dˆ-2.1. The area of a floe scales like Dˆ2. So the area distribution a(D) should scale like Dˆ(-2.1 + 2) = Dˆ-0.1. That means the area distribution should DECREASE as D increases from D=0.2 to D=3. But Figure 3b shows a(D) increasing as D increases over that range. Is there something wrong with the plots, or with my analysis?

Page 5, line 13. "the majority of the large floes are composed of two or more welded

pancakes." Does this explain some of the scatter in Figure 3a, in which a welded pair of pancakes would have a large major axis (D1) compared to minor axis (D2)?

Page 6, lines 16-18. It's commendable that the authors applied a goodness-of-fit test, and that they admit that neither the small nor the large floe size regime follows a power-law distribution according to that test. This result is actually not too surprising, given the very small size ranges over which the power laws were fit.

Page 6, lines 19-20. "N(D) possesses a slightly concave-down curvature across all the diameter ranges (in a log-log plane)". This phenomenon has been noted, or can be seen, in many previous studies, such as Rothrock and Thorndike 1984, Toyota et al 2016, Wang et al 2016 (JGR), and Stern et al 2018, where it is discussed at length.

Page 6, lines 25-29. This is a good paragraph, with entirely appropriate conclusions. Please note that it applies only to the pancake floes analyzed here, and not to the FSD in general.

Page 6, Conclusions. This section simply re-hashes the division of the FSD into three regimes. It claims that the small and large floe size regimes are "qualitatively close to power law scalings", but that is a very dubious characterization, especially for the large floe regime, where the range of floe sizes spans less than half an order of magnitude: $\log(10.8/4) < 1/2$.

The authors do not give a mechanism by which the FSD of the small floe regime comes to be qualitatively close to power-law scaling. They only state that the "pancakes are experiencing thermodynamic growth". How does that lead to power-law scaling?

For the large floe regime, they write that "floes are typically formed by welding". That's a mechanism that can be easily simulated in a numerical experiment, and the results compared to the actual (observed) distribution. I have taken the liberty of conducting such an experiment, which did not take very much time to code up – see the attached figure. I started with 20,000 floes whose sizes were distributed according to a power

law with exponent -2 and ranging from 0.25 to 3.0 meters in diameter. See the black curve in the attached figure. I then simulated a welding process in which two randomly chosen floes were welded together according to $D\_new = sqrt(D1^2 + D2^2)$ where D1 and D2 are the diameters of the floes to be welded, and D_new is the diameter of the welded floe. The floes with D1 and D2 are removed from the distribution, D_new is added to the distribution, and the process is repeated 5000 times, leaving 15,000 floes. The resulting distribution is shown by the red curve. The procedure is repeated again (5000 times), leaving 10,000 floes (green curve), and again (5000 times), leaving 5000 floes (blue curve). The blue curve has some qualitative similarities to Figure 3d. This is not a very sophisticated simulation, and I am not suggesting that the authors need to do something like this, but it does demonstrate the potential for mimicking certain processes. Of course a physical model would be better, but that is probably beyond the scope of the present study.
* * *
[Figure]

[Figure]

**Fig. 1.**

---

## Author Comment (AC1) · 15 Oct 2018

CH: This manuscript uses shipboard camera images taken over a transect of the Antarctic MIZ to describe the pancake floe size distribution in that region. The manuscript is sets out its goals, and accomplishes them concisely and straightforwardly, and so I recommend its publication in short order: this information is valuable and interesting to those who are trying to evaluate and understand sea ice models that incorporate the physics of small floes. I do have some relatively minor issues, mainly related to presentation and data processing, that I would like to see improved upon before publication. These are listed below - if any comments are unclear please contact

[Figure]

me with questions.

AA: We thank the referee for his very positive comments. Below we provide answers to all the comments.

CH: Page 1 Line 10 - I believe the paper of yours truly you mean to cite is H+T 2015, as that is the model paper.

AA: We agree that H+T 2015 is the better reference and thus we added it in the revised manuscript.

CH: Page 1 Line 11 - While it is true early results show the importance of the FSD at the edge, only floe breaking by waves and floe melting has been put in models, so it isn't true that this is the place where floes are most important as we don't have a handle on their evolution deeper into the pack.

AA: We have changed the statement.

CH: Page 1 Line 12 - Please cite either Steele, 1992, or Horvat et al, 2016 when making the statement about floe melting.

AA: Reference to Steele 1992 has been added.

CH: Page 1 Line 13 - I would delete "formed ... currents" as you also mention the importance of waves for pancake formation, and winds and currents can significantly alter the formation mechanism of sea ice in the MIZ.

AA: The phrase has been deleted.

CH: Page 1 Line 15 - change "exhibits" to "resembles" as you argue below the inapplicability of fractal scaling.

AA: The change has been made.

CH: Page 2 Line 3 - add citations to Herman 2014, 2017 here as it is important for readers to know there is really not much evidence of a multi-decadal power law.

[Figure]

AA: The section has been rewritten and clarified to provide a better overview of previous literature, including the papers by Herman (noting that Herman 2014 should be Herman 2010).

CH: Page 2 Line 5 - Both papers cited here really argue *against* the adoption of FSD power laws, not for them!

AA: We agree with the reviewer. We rephrased this section and added a reference to Herman et al. 2017 to make this statement clearer.

CH: Page 2 Line 16 - "the *Antarctic* sea ice annual mass budget".

AA: Added.

CH: Page 2 Line 18 - I'm not sure the statement about pancakes being more common is supported by the Roach et al paper as it is a point measurement.

AA: We changed "common" to "frequent than in the past". We added a reference to Wadhams et al. (2018) where such a claim is made explicitly. We have kept the reference Roach et al. (2018), which specifically focusses on pancakes and where this statement, supported by a number of references, appears at the end of the first paragraph of the Introduction.

CH: Page 3 Fig 1 - I would like to see this visualization improved, and the caption more descriptive. For example, it isn't clear that (a-b) and (c) are on different axes immediately, and isn't mentioned that (c) is the cutout in (a-b). Also, are the measurements you are making the dark line in (c)? Could you add the green dash to (a-b) as well?

AA: We now indicate explicitly in the caption that (c) is a subdomain of (a) and (b) where it is indicated by a white frame. We also added that the black part of the track in (c) indicates where cameras were operational and measurements undertaken. The green mark in (a) and (b) is superfluous as the area of interest is already framed in white and would clutter the figure.

CH: Page 3 Line 8 - explain what you mean about statistical independence.

AA: We now added that this means that the sampled area in two subsequent images is different (i.e. no overlap) thus all the floes are only measured once.

CH: Page 3 Line 8-15 - generally, please explain the operation used to compute the pancakes as this is very important information for reproducing or building from this work.

AA: A brief description has been added. The procedure is fairly standard and only uses a series of MatLab built-in functions. We point out that development of the algorithm is not the focus of this communication. Details are now given in the supplementary material. The algorithm and data are available upon request.

CH: Page 4 Line 5 - please explain why one might decline an image.

AA: Visual inspection of the processed images quickly reveals images to be disregarded, e.g. when the reconstructed floes didn't match the greyscale image in a macroscopic way. In general, images where discarded when a large number of separated floes were merged together (i.e. only one identified floe by the algorithm, we define such instance artificial welding) or when single floes were split in a number of smaller floes by the algorithm.

CH: Page 4 Line 8 you explain why AMSR might *overestimate* concentration, but for many points along the track it underestimates concentration - explain.

AA: AMSR2 averages two daily swaths. During the measurements the intense storm conditions induced an ice drift of the order of 100km Eastward. In this circumstance it is likely than one or even both swaths occurred over open water thus leading to underestimation the instantaneous observed ice concentration (e.g. at the beginning of the recording). A comment has been added in the revised manuscript.

CH: Line 18 - "prone to error" - what kind of error? why?

AA: The extent of these floes is only few pixels. Higher resolution (i.e. px/m) would be required to reliably reconstruct the shape of these floes. A comment has been added in the revised manuscript.

CH: Line 19 - why were the welded floes excluded? Isn't this the process by which these pancakes are said to form? What criteria is used to pick floes to exclude, and how does this affect the tail of your distributions in Fig 3?

AA: Only artificially welded floes are excluded, i.e. those that are made by two separated floes but the algorithm returns as one single floe. Real welded floes are still included. A comment has been added in the revised manuscript to clarify the concept of artificially welded floes.

CH: Line 20 - area = 1.55 kmЁЄ2 - I thought the swath was 28 m, which would mean you traveled 55 km into the ice, not close to 100 km.

AA: The swath itself would be 55km by 28m but this is distributed over a ship track of 100km. Two subsequent images are not contiguous in space to avoid overlap and guarantee statistic independence of the sampled floes. A comment has been added in the revised manuscript.

CH: Page 5 Figure 3 - You have extra space in this figure - could you please also plot the area weighted FSD rather than the number size distribution? The area size distribution is what appears in the Roach et al model and so would be good to see. You can estimate how much spread there is in (a) by taking $D\_1$ and adding white noise to it and calling that $D\_2$, then re-sorting in the instances $D\_2 > D\_1$. the magnitude of the white noise that is required to get the fit line would tell you how much error there is in assuming a circle.

AA: Figure 3 has been revised, with additional subplots and insets. The area distribution is presented in 3b (cumulative) and 3e (pdf, previously in 3b).

In response to this comment we show the empirical distribution of the D2/D1 ratio in
figure 3a which provides information on the scatter of D2/D1. To better describe the shape of the floes we added the scatter plot of the circularity as a function of the diameter (figure 3d in the revised manuscript). We believe this to be a better way to describe the shape of the floes and is more commonly used than the method suggested by the referee.

CH: Line 2 - why the mode = peak probability? Why not the mean, and could you report the "roundness" of the floes?

AA: The mode is extracted from the floe area distribution (i.e. the FSD expressed in terms of area instead of floe number as shown in Fig 3b and 3e in the revised manuscript). Mean and median are also reported in the revised manuscript noting that in terms of area distribution mode, median and mean are all about 3.1m.

CH: Line 4 - That the probability of exceedence hides the fact of a non-power-law distribution is extremely interesting and while some have discussed this in model papers, to my knowledge this is the first time this has been evidenced *on purpose* in an observational paper. I would like to see this highlighted!

AA: We added a statement to highlight this concept in the conclusions.

CH: Line 12 - I don't think you can say that there is a different physical mechanism to make larger floes as you only have a point observation of welding.

AA: Visual inspection of a large number of the images reveals that a considerable number of the larger floes are formed by smaller floes welded together. The line of welding is clearly visible from the images (see video in the supplementary material). These relatively large floes contribute the most in shaping the tail of the FSD. It is very likely then that welding process (for which importance has been shown by Roach et al., 2018), is the physical process that mostly affects the FSD in this regime, but this conjecture has to be further verified. The statement in the manuscript has been modified.

CH: Line 25 - The point about dropping a priori assumptions is good, I would add that using a KDE is in some ways equivalent to fitting distributions to power laws: both are not derived from first principles and so both give little insight into the actual physics governing the distribution.

AA: We made no change in response to this comment, as the point we were trying to make is that using a KDE involves no assumptions about the shape of the FSD.

Please also note the supplement to this comment:
https://www.the-cryosphere-discuss.net/tc-2018-155/tc-2018-155-AC1-supplement.zip

[Figure]

[Figure]

**Fig. 1.**

---

## Author Comment (AC2) · 15 Oct 2018

R2A: The authors acquired images of pancake ice floes from a ship-based camera on July 4, 2017, in the Antarctic marginal ice zone at about 30E, 62S. An automatic algorithm identified pancake ice floes in the images, whose size distribution was then plotted. They found three size regimes: diameters 0.25 to 2.3 meters, 2.3 to 4 meters, and 4 to 10 meters (see Figures 3b and 3d). They conjecture that the small regime is driven by the growth of pancakes from frazil ice, and the large regime is driven by the welding together of pancakes.

AA: We thank the reviewer for his/her comments and constructive criticisms. The

manuscript has been modified to address the comments and detailed answer to each of the comments is given below.

R2A: This paper is basically a report of data analysis. The conjectures regarding the small and large floe regimes are just that – conjectures – without supporting evidence. I have questions about the analysis, detailed below, as well as other comments. In my opinion, this paper needs major revisions.

AA: We agree with the reviewer that this brief communication reports data analysis, however, this overlooks that the present work increases knowledge on the subject as it provides the first quantitative measurements of pancake ice floes in the Antarctic marginal ice zone, as well as the first assessment of the pancake ice floe distributions (for area and diameter), based on a very large number of floes (results are statistically significant). We also compare with distributions traditionally proposed for sea ice and assess floe shape. Moreover, we point out that our conjectures (as we ourselves define them) need to be verified, and that they are based on previous work of Roach et al. (2018), in which these mechanisms are studied but without discussing the pancake ice floe size distribution and shape.

R2A: Page 1, lines 9-10. The floe size distribution (FSD) was first integrated into a sea-ice model by Zhang et al in 2015 and 2016.

AA: We added references to Zhang et al. (2015,2016).

R2A: Page 1, lines 12-15. The authors imply that there is not much "field data" available on floe sizes. I assume this refers to in-situ data such as that acquired from a ship in the ice. But there is plenty of remote sensing data, and it's not clear to me that field data is any better than remote sensing data, so the lack of field data does not seem like a shortcoming. The only advantage I can see to field data is the higher spatial resolution – in this case, the ability to identify floes as small as 0.25 m in diameter. But this advantage is not mentioned by the authors. Perhaps one of the values of this study is that it identifies floes that are smaller than in any other study. The fact that it consists

of data collected in the field is not in itself a selling point, in my opinion.

AA: We understand the possible misunderstanding with the use of "field data". We have clarified the different role played by in situ observations of smaller floes in the Southern Ocean and that there are no previous observations of pancake ice in this region, as their small scale makes them difficult to resolves from space.

R2A: Page 2, lines 2-4. This is really an oversimplification. It is certainly true that observations do not support a unique scaling exponent of the FSD – that was the subject of an entire paper (Stern et al 2018) which is cited by the authors later but not here. Only some of the 18 studies examined in that paper report "two distinct scaling exponents". The authors imply that those exponents are given by Toyota et al 2011, but other studies have found different exponents over different ranges, and some have reported that a single exponent characterizes the FSD.

AA: We rephrased the sentence to clarify this concept, also using Stern et al. (2018) as the main reference. We note that no previous works refer to pancakes, and we have further emphasized this as a novelty of our study.

R2A: Page 2, lines 5-6. "The validity of power law scaling has not been demonstrated yet and its adoption is mostly justified by the wide range of floes diameters". Actually the validity of power-law scaling has been demonstrated in some cases, and I have never seen a paper that claims that power-law scaling is justified by the wide range of floe diameters. The papers cited by the authors don't make that claim.

AA: We thank the reviewer for helping us to clarify this point. We rephrased the section to emphasise that the alpha = 2 exponent is not universal, and the power law behaviour is not verified in all the cases reported in the literature. We also added references to Horvat & Tziperman (2017) and Herman et al. (2017) to strengthen the statement.

R2A: Page 2, lines 6-7. "Scaling parameters are typically estimated on the log-log plane with a least square fit" – it would be good to note that such a procedure leads to
a biased estimate of the scaling parameter.

AA: This note has been added.

R2A: Page 2, line 12. "Existing observations do not provide quantitative descriptions of the floe size distribution for pancake ice floes", but line 19 says "Shen and Ackley (1991) reported pancake floe sizes..." so doesn't that contradict line 12?

AA: We have clarified that Shen & Ackley report a characteristic diameter only, i.e. not the floe size distribution as done in this work.

R2A: Page 2, line 26. "To our knowledge, the pancake floe size distribution has yet to be characterized." Take a look at: Parmiggiani, Moctezuma-Flores, and Guerrieri, Identifying pancake ice and computing pancake size distribution in aerial photographs, Proc. SPIE 10422, Remote Sensing of the Ocean, Sea Ice, Coastal Waters, and Large Water Regions 2017, 104220K (13 October 2017); doi: 10.1117/12.2277537

AA:. We rephrased the statement to include this reference. However, we note that the paper only focusses on development of a processing scheme for pancake floe detection. Only one image is analysed and results are reported in terms of pixels (no dimensions are reported), so the paper does not provide a quantitative characterisation of the pancake ice distribution, as we have done in this study over a larger region.

R2A: Section 2, Sea ice image acquisition. All good, nice work. I do have one comment: page 3 line 14 says that morphological image processing was used "to improve the shape of the pancake floes." I don't think "improve" is the right word! How about "to smooth"?

AA: Morphological operations are indeed used to improve or enhance the identification of the floes. We made no change in response to this comment in the manuscript but, to clarify the processing technique, we added a full list of morphological operations we adopted in the new supplementary material.

R2A: Page 4, lines 15-17. There is more than one way to define the floe diameter

D. The first study of the FSD, Rothrock and Thorndike 1984, used the mean caliper diameter.

AA: A note that the caliper diameter can be used as the characteristic dimension has been added.

R2A: Page 5, line 5. "a power law N(D) proportional to Dˆ-alpha as a benchmark and using the maximum likelihood method". Here, N(D) is the cumulative distribution function (CDF), but the maximum likelihood method yields the best-fitting exponent of the probability density function (PDF), not the CDF. If the authors used the maximum likelihood method to obtain the best-fitting exponent of the PDF (call it -beta), then they would have had to convert it to the exponent of the CDF (via -alpha = -beta + 1). Did they do this?

AA: We rephrased the statement to make clear that we used the procedure described in Stern et al. (2018), which is based on Clauset et al. (2009).

R2A: Page 5, lines 6-8. The authors note that the range of floe sizes for the large regime (4 to 10 meters) spans less than a factor of 10, and therefore "the estimation of the scaling exponent for D > 4 m is rigorously not applicable". Well, the same is true for the small floe regime (0.25 to 2.3 meters) – it spans less than a factor of 10, so apparently the estimation of the scaling exponent for D < 2.3 m is rigorously not applicable either. Doesn't that destroy the basis of the floe size analysis here?

AA: We rephrased to indicate that also in the small floe regime the range is less than a decade. However, this would only 'destroy' our analysis and findings if we were advocating that the distribution is a power law, when, in fact, we emphasise that it is not a power law.

R2A: Page 5, lines 12-13. "the steeper slope indicates that their size is governed by different underlying physical mechanisms." Or by the finite size effect, in which larger floes are under-observed because the finite size of the images makes it less likely to

see larger floes in their entirety. This has been described in the literature. Can the finite size effect be ruled out here?

AA: Indeed, finite size effects can be ruled out, as the floes for which the change in slope occurs (∼4 m) are considerably smaller than the image footprint (∼28 m x 28 m). Even the largest floe recorded (∼10 m) is less than 2 times than the dimension of the image. Therefore, we conjecture that the clipping at ∼10 m is due to a physical mechanism.

R2A: Page 5, Figure 3. It looks to me like Figure 3b (the area distribution, a(D)) is not compatible with Figure 3d (the PDF, n(D)). In 3b, the area distribution increases as D increases, from D=0.2 to D=3. In that range, alpha_S = 1.1 so alpha_S + 1 = 2.1 so the PDF n(D) scales like Dˆ-2.1. The area of a floe scales like Dˆ2. So the area distribution a(D) should scale like Dˆ(-2.1 + 2) = Dˆ-0.1. That means the area distribution should DECREASE as D increases from D=0.2 to D=3. But Figure 3b shows a(D) increasing as D increases over that range. Is there something wrong with the plots, or with my analysis?

AA: The reviewer's analysis is correct and we understand that the estimation of the co-efficients and their application to the area distribution require further explanation. We note that the value of the exponent depends on the range over which computations are made, and, in this regard, the value 2.3 m is arbitrary. Since the CDF is concave-down, by reducing the range over which the exponent is estimated a lower slope can be obtained (eventually the exponent might drop below alpha_S = 1, thus leading to an in-creasing area distribution). Moreover, the discrepancy between a power law behaviour for the number distribution and the measured area distribution further demonstrates that the underlying number distribution is, in fact, not a power law (as also indicated by the goodness of fit test). We now explicitly point out this apparent discrepancy: "We also note that an increasing alpha(D) in the small-floe regime (see Figs. 3e) is incon-sistent with a power law with alpha_S ≥ 1 and, thus, the area distribution confirms that the underlying number distribution is not a power law."

R2A: Page 5, line 13. "the majority of the large floes are composed of two or more welded, pancakes." Does this explain some of the scatter in Figure 3a, in which a welded pair of pancakes would have a large major axis (D1) compared to minor axis (D2)?

AA: We can't say with certainty if the scatter is due to welding. However, we note that the scatter is fairly homogeneous across all diameters in the range, whereas welding dominates for large pancakes only. This information was based on visual observations and we now provide a sample video of the acquired images as supplementary material to help the reader.

R2A: Page 6, lines 16-18. It's commendable that the authors applied a goodness-of-fit test, and that they admit that neither the small nor the large floe size regime follows a power law distribution according to that test. This result is actually not too surprising, given the very small size ranges over which the power laws were fit.

AA: We point out that the choice of fitting and testing a power law derives from the traditional sea-ice approach, well knowing that the range of diameters is small. A comment has been added in the revised manuscript.

R2A: Page 6, lines 19-20. "N(D) possesses a slightly concave-down curvature across all the diameter ranges (in a log-log plane)". This phenomenon has been noted, or can be seen, in many previous studies, such as Rothrock and Thorndike 1984, Toyota et al 2016, Wang et al 2016 (JGR), and Stern et al 2018, where it is discussed at length.

AA: As pointed out by Stern et al. (2018), this behaviour relates to the fact that the underlying distribution might be a truncated power law. We added a note on this in the revised manuscript.

R2A: Page 6, lines 25-29. This is a good paragraph, with entirely appropriate conclusions. Please note that it applies only to the pancake floes analyzed here, and not to the FSD in general.

AA: We thank the reviewer for the positive comment and we added a note, as suggested.

R2A: Page 6, Conclusions. This section simply re-hashes the division of the FSD into three regimes. It claims that the small and large floe size regimes are "qualitatively close to power law scalings", but that is a very dubious characterization, especially for the large floe regime, where the range of floe sizes spans less than half an order of magnitude: $\log(10.8/4) < 1/2$.

AA: We rephrased and removed any reference to power-law behaviour. In the revised manuscript we write: "Two different behaviours are observed for smaller and larger pancakes on a log-log plane. The small-floe regime (D < 2.3 m), in which it is conjectured that pancakes are experiencing thermodynamic growth, is characterised by a mild negative slope (in terms of the floe number exceedance and probability density function), while the large-floe regime, in which floes are typically formed by welding (detected from visual analysis), is characterised by a much steeper slope noting that neither of the two regime conforms to a power law scaling."

R2A: The authors do not give a mechanism by which the FSD of the small floe regime comes to be qualitatively close to power-law scaling. They only state that the "pancakes are experiencing thermodynamic growth". How does that lead to power-law scaling?

AA: Based on observations by Roach et al. (2018), we hypothesise that thermodynamic growth is a potential mechanism leading to the observed FSD in the small-floe regime. We rephrased the relevant section to clarify that this statement is fairly speculative and, in the spirit of a brief communication, only serves to stimulate further research. We certainly do not claim that this mechanism leads to a power law.

R2A: For the large floe regime, they write that "floes are typically formed by welding". That's a mechanism that can be easily simulated in a numerical experiment, and the results compared to the actual (observed) distribution. I have taken the liberty of conducting such an experiment, which did not take very much time to code up – see the

attached figure. I started with 20,000 floes whose sizes were distributed according to a power law with exponent -2 and ranging from 0.25 to 3.0 meters in diameter. See the black curve in the attached figure. I then simulated a welding process in which two randomly chosen floes were welded together according to D_new = sqrt(D1ËĘ2 + D2ËĘ2) where D1 and D2 are the diameters of the floes to be welded, and D_new is the diameter of the welded floe. The floes with D1 and D2 are removed from the distribution, D_new is added to the distribution, and the process is repeated 5000 times, leaving 15,000 floes. The resulting distribution is shown by the red curve. The procedure is repeated again (5000 times), leaving 10,000 floes (green curve), and again (5000 times), leaving 5000 floes (blue curve). The blue curve has some qualitative similarities to Figure 3d. This is not a very sophisticated simulation, and I am not suggesting that the authors need to do something like this, but it does demonstrate the potential for mimicking certain processes. Of course a physical model would be better, but that is probably beyond the scope of the present study

AA: The statements that large floes are formed by welding derive by visual analysis of the acquired images and observations made from the ship deck. We have added footage as supplementary material to support the statements. The reviewer's suggestion and proposed simulations are certainly interesting, and we also agree that a detailed investigation of this kind would go far beyond the scope of the present brief communication. We would like to emphasize the main aim of the work which is to report the first quantitative observations on the pancake ice floe size distribution and to stimulate further research.

In the revised manuscript we note that these results reflect observations collected under storm conditions (see Fig. 1) and thus they may not be applicable to all the sea-ice states. We indicate that simultaneous measurements of waves, floe size and heat fluxes under a number of different conditions are indeed needed to verify the conjecture that different physical mechanisms are responsible for the peculiar shape of the pancake ice floe size distribution with the intention of stimulating further research.

Please also note the supplement to this comment:
https://www.the-cryosphere-discuss.net/tc-2018-155/tc-2018-155-AC2-supplement.zip

---

## Author Response (AR2)

[revised manuscript text omitted]

*Editor Decision: Publish subject to minor revisions (review by editor) (06 Dec 2018) by Ted Maksym*

**Comments to the Author:**

*In my view, the author's have sufficiently addressed all the reviewer comments. Although reviewer #2 indicated major changes, it appears to me that most of the comments required minor changes to the text and have been adequately addressed.*

**Response to the Editor:**

We thank the Editor for the positive feedback.

***There is one comment, however, that I do not feel the author's adequately explained:***

*Page 5, Figure 3. It looks to me like Figure 3b (the area distribution, a(D)) is not compatible with Figure 3d (the PDF, n(D)). In 3b, the area distribution increases as D increases, from D=0.2 to D=3. In that range, alpha_S = 1.1 so alpha_S + 1 = 2.1 so the PDF n(D) scales like D^-2.1. The area of a floe scales like D^2. So the area distribution a(D) should scale like D^(-2.1 + 2) = D^-0.1. That means the area distribution should DECREASE as D increases from D=0.2 to D=3. But Figure 3b shows a(D) increasing as D increases over that range. Is there something wrong with the plots, or with my analysis?*

*The authors' response is that by changing the range over which alpha_S is determined, one could get different alpha_S because of the concavity of the area distribution. This is true (for example over D=1.5-3, one obviously), but does not explain why the a(D) increases monotonically from D~0.3-2.3 and the fit of alpha_S over that same interval results in alpha_S > 1. Based on the reviewer's analysis, it alpha_S should be less than 1 over this precise interval. I believe this may be an issue with how robust the fit in Figure 3f is. From D=1.5-2, n(D) is flat, so a fit over this range would have alpha+1 ~1, and this is consistent with an increasing a(D) based on the reviewer's analysis. But for D=0.3-1.5, alpha +1 looks to be slightly less steep than the red line fit in 3f, so alpha could be < 1 over this range as well, so that would also be consistent with a(D) increasing, although less steeply. That's all fine. But since the red line fit is to 0.3-2.3, and the blue line appears less steep than this over both the 0.3-1.5 and 1.5-2.3 ranges, why is the red line, which is fit over the full range 0.3-2.3 steeper than what you would intuitively expect for a fit over either of these ranges? Is it due to the fitting technique? If so, then it would seem that this fit is not particularly robust. The authors' response does not really answer why a(D) still increases over the full range of 0.3-2.3 yet the fit has alpha_S > 1.*

*Now what the exact value of alpha_S is is not so important for the main points of the paper, but this does suggest to me that there is a possible issue with the fit. This needs to be explained. The statement added to the text is too short, and doesn't really explain why the increase in a(D) is inconsistent with alpha_S>1 for the reader.*

**Response to the Editor:**

We use the power law fit, but the goodness of fit test reveals that the underlying distribution is not a power law.

In general a(D) ~ D^2 n(D), but we can only write a(D) ~ D^(-alpha+1) if the number distribution is a power law, and, as pointed out by the reviewer and the editor, we obtain an increasing a(D) only for alpha<1. However, the area distribution obtained assuming a power law provides inconsistent results because the underlying distribution is, in fact, not a power law. For example, we run few tests and the n(D) is qualitatively comparable to an exponential in the small floe regime, which is consistent with an increasing area. If we assume n(D)~exp(-D) we get a(D) increasing up to D=2, as a(D)~D^2exp(-D) has derivative (2-D)*D*exp(-D), which is positive for D<2.

Moreover, having assumed a power law and applied a maximum likelihood method over a limited range of diameters (i.e. less than a decade), we doubt the robustness of the computed exponent, which might drop below 1 (in the revised manuscript we write that the less than a decade interval "is not particularly meaningful or robust"). By applying a less rigorous least square fitting, we find alpha ~ 0.9, which is consistent with growing area distribution. Indeed, the qualitative fit does not vary dramatically for alpha values between 0.9 and 1.1 (see below).

[Figure]

*Minor edits/clarifications:*

*Line 2 change to "made up of floes*

Changed

*Abstract – Based on the comments and your conclusions, I wonder if it is worth stating in the abstract that the FSD does not exhibit power law behavior?*

Added

*Page 2, Line 5 – are you sure that "most" of the observations conform to a truncated power law? I believe Stern et al (2018) only states that several studies show this, , or if only some conform to a truncated power law. Also, my understanding is that the "split power law" could be an artifact of this truncation (or finite size effects) – see figure 3 of Stern et al (2018). Your text as written could be read to imply that previous observations show either a*

*truncated power law or split power law. But I believe many of the prior studies only suggest a single power law. Please clarify this statement.*

As stated by Stern (2018), while the truncated power law fits most of the previous datasets, Stern (2018) does not exclude that in certain cases the split power law might consistent with the data (i.e. there is physical basis for the split power law).
In particular, the truncated power law cannot describe the data of Steer (2008) in which the probability density function is shown, as discussed in Stern (2018) in 4.1 at the end of the first paragraph. We clarified these issues.

*Page 2, Line 11-12 – despite your response to the reviewer, you still state that the power law scaling has not been demonstrated, and you still make the claim that the power-law scaling is justified by the wide range of diameters. Please modify the former to state that the power law has been demonstrated for only some cases, and the latter to state something more reasonable – isn't it justified simply because it appears to follow a power-law, even if this is not rigorously verified?*

We use "demonstrate" when it is mathematically proven. We agree that Stern has shown the truncated power law fits most of the previous observations, but this doesn't constitute a proof. We write that the power law has been verified for most cases, but its universality has not been demonstrated yet.

*Figure 1, caption – "Environmental conditions"*

*Figure 1, caption – "sourced from the AMSR2"*

*Figure 1, caption – "The black dots denote"*

*Page 4, line 17 – "image processing"*

*Page 4, line 19 – perhaps to better address the reviewer's comment, this could be changed to "image processing to improve the fidelity of the shape of identified pancake floes"*

*Page 4, line 19 – "light and ice conditions"*

*Figure 3 caption – "floe diameter with two power laws", "(D > 4m), respectively". Also I think it would be better if the panels were described in order (i.e. a, b, c, d, e, and f).*

*Page 5, line 7 – be better to state that this could result in an over- or underestimation of the in situ ice concentration. There is no reason to expect this average would always cause an underestimation.*

*Page 6, line 28 – "transition between these regimes"*

All changed.

*Page 7, line 12-13 – are you sure you can say estimation of the scaling exponent it is not rigorously applicable because of the less than a decade of length scales? I am not certain, but I would guess that for a large amount of well-behaved observations, one could achieve a*

*rigorous fit (at least over that less-than-decade range). My point is "rigorously" might not be a precise word here. Perhaps better to say "not particularly meaningful"? Leave alone if it is correct.*

Changed. We also added that it is probably not robust.

*References – please ensure your references are consistent with The Cryosphere style. For instance, you have full journal titles, whereas journal style is to abbreviate.*

Changed. We used bibtex in the Cryosphere style.

*Page 9 line 7 – should annals be capitalized here?*

*Page 9, line 17 – Reports should be capitalized.*

*Page 9, line 20 – Herman et al. is now in The Cryopshere (https://www.the-cryosphere.net/12/685/2018/), so this reference should be updated.*

*Supplementary material – this supplement is quite short, so could this be better included as an appendix?*

All changed.